# SARS-CoV-2 Infection of Unvaccinated Liver- and Kidney-Transplant Recipients: A Single-Center Experience of 103 Consecutive Cases

Hailey Hardgrave [1,2], Allison Wells [2,3], Joseph Nigh [2,3], Tamara Osborn [2], Garrett Klutts [2], Derek Krinock [2], Mary Katherine Rude [4], Sushma Bhusal [5], Lyle Burdine [2,3] and Emmanouil Giorgakis [2,3,*]

1   College of Medicine, University of Arkansas for Medical Sciences, Little Rock, AR 72205, USA; hjhardgrave@uams.edu
2   Department of Surgery, University of Arkansas for Medical Sciences, Little Rock, AR 72205, USA; awells2@uams.edu (A.W.); jnigh@uams.edu (J.N.); tosborn@uams.edu (T.O.); gklutts@uams.edu (G.K.); dkrinock@uams.edu (D.K.); lburdine@uams.edu (L.B.)
3   Division of Solid Organ Transplantation, University of Arkansas for Medical Sciences, 4301 W Markham St, Little Rock, AR 72205, USA
4   Division of Gastroenterology and Hepatology, University of Arkansas for Medical Sciences, Little Rock, AR 72205, USA; mkrude@uams.edu
5   Division of Nephrology, University of Arkansas for Medical Sciences, Little Rock, AR 72205, USA; sbhusal@uams.edu
*   Correspondence: egiorgakis@uams.edu; Tel.: +1-501-526-6390

**Abstract:** Severe Acute Respiratory Syndrome Coronavirus-2 (SARS-CoV-2) was declared a pandemic in March 2020. Its reported impact on solid-organ-transplant-recipient morbidity and mortality has varied. The aim of this study was to present the effect of transplant status, patient comorbidities and immunosuppression modality on the survival of solid-organ-transplant recipients who contracted SAR-CoV-2 during the pre-vaccination era, at a single academic transplant center. Patients (n = 103) were assessed for 90-day mortality. A univariate analysis identified an age of over 60 years (HR = 10, $p = 0.0034$), Belatacept (HR = 6.1, $p = 0.022$), and Cyclosporine (HR = 6.1, $p = 0.0089$) as significant mortality risk factors; Tacrolimus was protective (HR = 0.23, $p = 0.022$). Common metabolic comorbidities (hypertension, diabetes, obesity) did not stand out as risk factors in our patient cohort. This study on the unvaccinated is expected to facilitate a paired comparison of outcomes in transplanted patients who contracted SARS-CoV-2 during the latter period of the pandemic, when broad SARS-CoV-2 vaccination and novel antibody treatments became broadly available.

**Keywords:** solid-organ transplant; COVID-19; unvaccinated; immunosuppression

## 1. Introduction

In January 2020, a novel coronavirus now known as Severe Acute Respiratory Syndrome Coronavirus-2 (SARS-CoV-2) was first identified in Wuhan City, China [1]. The World Health Organization announced SARS-CoV-2 as a Public Health Emergency Concern and declared the viral outbreak a pandemic in March 2020 [2]. Exactly two years since, the U.S. has had more than 79 million confirmed SARS-CoV-2 cases and almost one million fatalities [3]. During the same period, over 452 million cases and 6 million SARS-CoV-2-related deaths have been reported globally [4].

As has been previously discussed by this research group and others, individuals that have received liver and kidney transplants are at a significantly heightened risk for morbidity and mortality from SARS-CoV-2 infection compared to the general population [5–14]. Liver- and kidney-transplant recipients have higher rates of diabetes, obesity, hypertension, and cardiovascular disease, which have all been identified as risk factors for severe SARS-CoV-2 complications in early reports at our institution and by others [6,8,11,15–18].

Early anecdotal experience [8] and later reports have reported a higher mortality risk among kidney-transplant recipients following SARS-CoV-2 infection compared to the liver-transplant recipients [19], with both groups having higher hospitalization and intensive-care-unit-admission rates. These early observations were debated in the later periods of the pandemic. This perhaps reflects the higher quality of care and closer surveillance of the transplant patients compared to the general population as well as the better understanding of the disease pathophysiology and effective treatments as the pandemic evolved, among other reasons [6,7,9,11,12,14,15,17,20,21].

Despite the plethora of published reports, the role of immunosuppression in SARS-CoV-2 severity in post-transplant patients remains unclear: Standard immunosuppression could potentially suppress the immune system's capacity to mount a sufficient response to neutralize the viral insult, modulate systemic inflammatory storm, or suppress viral replication [5,6,9,11,13,15,20–27]. By convention, most transplant clinicians modify the maintenance of immunosuppression in transplant patients infected by SARS-CoV-2, frequently by decreasing or even discontinuing antimetabolites [6,12,13,15,23,24,27]. The international society for heart and lung transplantation has officially recommended consideration for using mycophenolate mofetil, mTOR inhibitors, and azathioprine in transplant patients with moderate to severe SARS-CoV-2 [28]. Virus-targeted immunotherapies, i.e., monoclonal antibodies (MABs) and convalescent plasma have emerged as potential treatments. Studies have reported a decrease in hospitalization need and mortality rates following the use of MABs in high-risk groups, such as the immunocompromised transplant recipients [29–31].

This study aimed to study the SARS-CoV-2-specific mortality and associated risk factors of a cohort of 103 consecutive unvaccinated solid-organ-transplant recipients that were transplanted at a single academic transplant center, using a prospectively populated institutional SARS-CoV-2 transplant registry.

## 2. Materials and Methods

### 2.1. Study Inclusion

At the onset of the pandemic, we sought to build and populate a registry of all transplant recipients who contracted the disease, after obtaining Institutional Review Board exemption [8]. The study included all consecutive adult solid-organ-transplant recipients 18 years of age or above who had previously received a solid-organ transplant in our institution (liver, kidney or both) and tested positive for SARS-CoV-2 between 1 February 2020 and 18 February 2021. Subjects were included regardless of the elapsed time between transplantation and the positive SARS-CoV-2 test. All patients had functioning grafts at the time of enrollment. A positive SARS-CoV-2 diagnosis was determined via either a positive polymerase chain reaction or a positive antigen test [9]. The subjects were either completely unvaccinated or less than 2 weeks from their last vaccination.

### 2.2. Database Creation

As already described in our preliminary reports, an institutional Research Electronic Data Capture database was created, populated by all consecutive eligible de-identified subjects [8,18]. The collected data included patient demographic characteristics, comorbidities, transplant details, immunosuppression regimen, and SARS-CoV-2-specific treatment and outcomes [8,18]. Patients were followed for a 90-day period from the time of diagnosis [8,18].

### 2.3. Statistical Analysis

Subjects were divided into groups of survivors and fatalities at the end of the 90-day follow-up period. Categorical variables were reported as the number and percentage of the total group (%) and compared using the Fisher's exact test [32,33]. Continuous variables were reported as a median and interquartile range (lower quartile, upper quartile) and compared using the Wilcoxon rank sum test [8,32,33]. A univariate Cox regression model

was performed on the above-discussed variables and a Kaplan–Meier survival curve was constructed by age group [32].

## 3. Results

A total of 103 patients were enrolled, with 76 kidney-transplants recipients, 23 liver-transplant recipients, and 4 simultaneous liver–kidney-transplant (SLK) patients. There was a total of 10 90-day mortalities and 93 surviving patients. Patient demographic information, transplant type, comorbidities, and immunosuppression-regimen descriptions are shown in Table 1. Age, gender, transplant type, and comorbidities were statistically similar between the groups. There was a statistically significant difference ($p < 0.001$) between the median age of 67 and 52 in the dead and survivor groups, respectively. Significant differences also existed between groups in terms of immunosuppression regimens, namely Tacrolimus ($p = 0.037$) and Cyclosporine ($p = 0.029$).

**Table 1.** Patient Demographics, Transplant Type, Comorbidities, and Immunosuppression. Age reported as Median (IQR); analyzed with Wilcoxon rank sum test. Categorical variables reported as n (%); analyzed with Fisher's exact test.

|  | **Deaths** **N = 10 (%)** | **Survivors** **N = 93 (%)** | **Total** **N = 103 (%)** | **Mortality** **Rate (%)** | ***p* Value** |
|---|---|---|---|---|---|
| **Age** | **67 (62, 70)** | **52 (42, 59)** | **54 (42, 62)** |  | **<0.001** |
| **Gender** |  |  |  |  | >0.900 |
| Male | 6 (60.0) | 52 (56.0) | 58 (56.3) | 10.3 |  |
| Female | 4 (40.0) | 41 (44.0) | 45 (43.7) | 8.9 |  |
| **Transplant Type** |  |  |  |  | 0.600 |
| Liver | 1 (10.0) | 22 (24.0) | 23 (22.3) | 4.3 |  |
| Kidney | 9 (90.0) | 67 (72.0) | 76 (73.8) | 11.8 |  |
| SLK | 0 | 4 (4.3) | 4 (3.9) | 0.0 |  |
| Total | 10 | 93 | 103 | 9.7 |  |
| **Comorbidities** |  |  |  |  |  |
| HTN | 10 (100.0) | 69 (74.0) | 79 (76.7) | 12.7 | 0.110 |
| Diabetes | 7 (70.0) | 37 (40.0) | 44 (42.7) | 15.9 | 0.094 |
| Obesity | 0 (0) | 16 (17.2) | 16 (15.5) | 0 | 0.354 |
| Coronary Artery Disease | 2 (20.0) | 8 (8.6) | 10 (9.7)) | 20.0 | 0.250 |
| **Immunosuppression** |  |  |  |  |  |
| Tacrolimus | 6 (60.0) | 82 (88.0) | 88 (85.4) | 6.8 | 0.037 |
| Cyclosporine | 3 (30.0) | 5 (5.4) | 8 (7.8) | 37.5 | 0.029 |
| Prednisone | 7 (70.0) | 48 (52.0) | 55 (54.4) | 12.7 | 0.300 |
| MMF | 7 (70.0) | 66 (71.0) | 77 (70.9) | 9.1 | >0.900 |
| Sirolimus | 1 (10.0) | 5 (5.4) | 6 (5.8) | 16.7 | 0.500 |
| Belatacept | 2 (20.0) | 3 (3.2) | 5 (4.9) | 40.0 | 0.073 |
| Azathioprine | 0 | 3 (3.2) | 3 (2.9) | 0.0 | >0.900 |

SLK, simultaneous-liver kidney transplant; MMF, mycophenolate mofetil.

A univariate Cox regression model was performed for ages greater than 60 and immunosuppression regimen, shown in Table 2. Patients aged >60 were associated with a higher hazard ratio (HR) (HR = 10, $p = 0.0034$), as well as Cyclosporine (HR = 6.1, $p = 0.0089$) or Belatacept for the immunosuppression maintenance (HR = 6.1, $p = 0.022$), contrary to Tacrolimus (HR = 0.23, $p = 0.022$). No significant mortality risk or benefit was seen in patients taking prednisone, MMF, Sirolimus, or Azathioprine.

**Table 2.** Univariate Cox Regression Model of Selected Variables.

|  | Beta | HR | 95% CI | *p* Value |
|---|---|---|---|---|
| **Age > 60** | **2.30** | **10.00** | **(2.10–48.00)** | **0.003** |
| **Immunosuppression** |  |  |  |  |
| Tacrolimus | −1.50 | 0.23 | (0.06–0.81) | 0.022 |
| Cyclosporine | 1.80 | 6.10 | (1.60–24.00) | 0.009 |
| Prednisone | 0.72 | 2.10 | (0.53–7.90) | 0.300 |
| MMF | −0.05 | 0.95 | (0.25–3.70) | 0.950 |
| Sirolimus | 0.59 | 1.80 | (0.23–14.00) | 0.570 |
| Belatacept | 1.80 | 6.10 | (1.30–29.00) | 0.022 |
| Azathioprine | −17.00 | $3.90 \times 10^{-8}$ | (0-Inf) | 1.000 |

MMF, mycophenolate mofetil.

A Kaplan–Meier survival curve and the associated life table are shown in Figure 1 and Table 3, respectively. No SARS-CoV-2-related deaths within 90 days post-infection occurred in the youngest (20–51) age group. For the rest of the groups, deaths occurred 2 to 45 days post-SARS-CoV-2 diagnosis. The oldest patient group (aged $\geq$ 72) had the least survival probability (75%) compared to the rest (reference 20–51 years; $p < 0.001$).

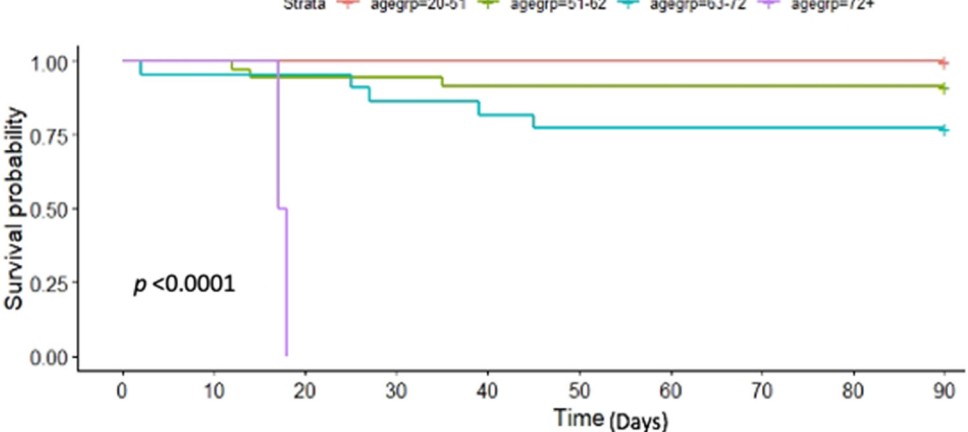

**Figure 1.** Kaplan-Meier survival curve of unvaccinated SARS-CoV-2 positive solid organ transplant recipients, stratified by age groups (years): 20–51, 51–63, 63–72, >72. Patient survival was inferior in the oldest age group ($p < 0.0001$).

**Table 3.** Life Table by Age Group.

|  | Number at Risk | | | |
|---|---|---|---|---|
| **Age Group (years)** | **0-Days** | **30-Days** | **60-Days** | **90-Days** |
| 21–51 years | 44 | 44 | 44 | 44 |
| 51–62 years | 35 | 33 | 32 | 32 |
| 63–72 years | 22 | 19 | 17 | 17 |
| 72+ years | 2 | 0 | 0 | 0 |
|  | Survival (%) | | | |
| **Age Group (years)** | **0-Days** | **30-Days** | **60-Days** | **90-Days** |
| 21–51 years | 100 | 100 | 100 | 100 |
| 51–62 years | 100 | 94.29 | 91.43 | 91.43 |
| 63–72 years | 100 | 86.36 | 77.27 | 77.27 |
| 72+ years | 100 | 0 | 0 | 0 |

## 4. Discussion

During the study period, 103 solid-organ-transplant patients were diagnosed with SARS-CoV-2 at our institution, with a 9.7% SARS-CoV-2-specific mortality rate within three months of diagnosis. This finding was similar to our early institutional experience and to reports by others during the first year of the pandemic, before vaccinations had

become broadly available [7–9,11,17,18]. In our cohort, most of the infected patients were kidney-transplant recipients, which aligned with the higher prevalence of this transplant subgroup. Similar to our preliminary reports [8,18], the kidney-transplant-recipient SARS-CoV-2 mortality rate was 11.8% vs. 4.3% among the liver-transplant recipients, with a calculated relative risk of 2.7 (95% CI 0.36–20.3). There were no reported deaths among the 4 combined liver–kidney-transplant recipients who had tested positive for SARS-CoV-2.

Hypertension and diabetes were present in 12.7% and 15.9% of the deaths. Despite early reports by others, these comorbidities were not associated with increased mortality in our cohort. SARS-CoV-2 mortality increased with advancing age, a finding described in general population outcomes [34].

Mirroring the practice of decreasing or discontinuing MMF in the presence of a viral infection, such as Cytomegalovirus, the antimetabolite dose was decreased or held for two weeks from the time of SARS-CoV-2 diagnosis. Our study failed to demonstrate any significant MMF effect on SRS-CoV-2-related mortality. However, more than 60% of patients who were taking MMF at the time of diagnosis had this medication held or decreased.

In our patient cohort, Tacrolimus demonstrated a protective effect (HR = 6.1, $p$ = 0.022), an observation already reported by others [35]. A meta-analysis of 11 cohort studies investigating the impact of immunosuppression on SARS-CoV-2 suggested that Tacrolimus usage did not impact mortality or SARS-CoV-2 infection severity [36]. In our cohort, only eight (7.8%) patients had been on Cyclosporine at the time of the SARS-CoV-2 infection, three of whom died. In the univariate Cox regression, Cyclosporine was associated with a 6.1 (95% CI 1.6–24) death risk, contrary to a favorable 0.23 (95% CI 0.064–0.81) when using Tacrolimus as a Calcineurin inhibitor. These findings do not necessarily imply causation and should therefore be interpreted with caution; the findings may be attributed to the small patient sample and/or lack of control of confounding variables, including, but not limited to, the underlying indication for the switch to Cyclosporine from Tacrolimus, which has been the standard of care in our institution.

Two out of five (60%) patients who had been on Belatacept at the time of SARS-CoV-2 diagnosis eventually succumbed to the disease (HR = 6.1, $p$ = 0.022). The literature is largely limited to case studies on the impact of Belatacept on SARS-CoV-2 outcomes. As a T-cell co-stimulation inhibitor, Belatacept is theorized as a potential mitigator of the cytokine storm caused by SARS-CoV-2 infection; however, it has also been shown to potentially increase the risk of severe opportunistic infections [37,38]. Similar to Cyclosporine, it remains unclear if this apparent positive correlation of Belatacept with severe SARS-CoV-2 infection reflects causation; since Belatacept is a choice often reserved for patients intolerant to CNIs and/or with a significant cardiovascular burden or recent cardiac events, there might be confounders that have not been identified in this small population sample, such as the underlying indication of the patient being switched to Belatacept. Like in the case of Cyclosporine, it may be the underlying comorbidities that led to the immunosuppression-regimen switch rather than the immunosuppression choice *per se*, as the factors impacting the disease outcome.

As scientific evidence evolved along with the pandemic progression, treatment for SARS-CoV-2 for both inpatients and outpatients at this institution changed over the course of this study, in alignment with the federal guidelines and transplant organizations' recommendations. Monoclonal-antibody therapy was recommended for SARS-CoV-2-positive transplant recipients managed in the outpatient setting and became available near the end of the study period in December 2020. A total of 21 (20.38%) patients in this study received monoclonal-antibody therapy. Remdesivir and convalescent plasma were also used for inpatients meeting certain criteria. A total of nine (8.74%) patients received Remdesivir, and five (4.85%) received convalescent plasma. While the impact of these treatments was not analyzed as part of this portion of the study, it is reasonable to consider that their use may have mitigated the mortality in these patients, particularly towards the latter stages of the cohort, when antibody treatments became standardized and broadly available, particularly for the higher-risk subgroups.

These data were collected over a period when SARS-CoV-2 vaccination was not widely available, therefore providing an opportunity to assess the viral infection fatality in our immunosuppressed population prior to the broad implementation of SARS-CoV-2 vaccines.

Age cohorts stood out as remarkable predictors of outcome and provided for a more robust analysis. No patients died in the 20–51-year age group and patients in the >72-year group had the least survival probability (75%, $p < 0.001$). Other studies have found age to be one of the most important factors in predicting SARS-CoV-2 mortality. This study, combined with data from existing works, is perhaps suggestive of the need to provide more robust, earlier intervention in the older transplant population [39–42]. Novel treatments such as MABs, antiviral agents, and most importantly preventative measures, could prove particularly life-saving in this older group of unvaccinated SARS-CoV-2-positive transplant recipients.

Statistical limitations existed in this study due to the small sample size. This led to severe model instability when a multivariate Cox regression analysis was attempted, as well as some instability of the univariate regression model. Model instability is particularly prevalent in fields with zero covariates in the fatality group. An extension of this study is currently ongoing to capture a larger study population in the attempt to build a stable model for analysis.

## 5. Conclusions

Our SARS-CoV-2 transplant registry demonstrated an almost 10% death rate in the early pandemic era, when vaccinations were not yet available and MAB treatment options were still evolving. Despite a trend for the kidney-transplant recipients being more susceptible to severe disease, particularly at the outset of the pandemic, this did not reach significance, while age prevailed as the mortality predictor, increasing the death hazard by a factor of 10 over the age of 60. Tacrolimus immunomodulation was protective in our patient sample. However, these findings should be interpreted with caution, since they could be inherent to the well-known limitations of a small sample size and retrospective study bias. Randomized trials are needed to elucidate the various immunosuppression modalities' impact on disease progression. This pilot study, which was conducted in a highly endemic area of the disease and on a patient population with overall morbidity and mortality among the highest in the United States, may provide the control group for future high-quality propensity-score-matched studies.

**Author Contributions:** Conceptualization, M.K.R., S.B., L.B. and E.G.; Data curation, H.H., A.W. and E.G.; Investigation, H.H., J.N., T.O., G.K. and D.K.; Methodology, H.H. and E.G.; Project administration, E.G.; Resources, A.W., M.K.R., S.B., L.B. and E.G.; Supervision, E.G.; Writing–original draft, H.H., A.W. and E.G.; Writing–review & editing, H.H., A.W., J.N., T.O., G.K., D.K., M.K.R., S.B., L.B. and E.G.; Final revision and approval of the manuscript: E.G. All authors have read and agreed to the published version of the manuscript.

**Funding:** This research received no external funding.

**Institutional Review Board Statement:** The study was conducted in accordance with the Declaration of Helsinki, and approved by the Institutional Review Board (or Ethics Committee) of University of Arkansas for Medical Sciences (protoco l262269, 18 December 2020).

**Informed Consent Statement:** Patient consent was waived with approval from an institutional IRB as this was a chart review study with no direct patient contact, all collected patient data was deidentified, and no identifiable patient information is reported.

**Data Availability Statement:** Not applicable.

**Conflicts of Interest:** The authors declare no conflict of interest.

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
