# Peer review of "SARS-CoV-2 Infection of Unvaccinated Liver- and Kidney-Transplant Recipients: A Single-Center Experience of 103 Consecutive Cases"

_2673-3943, doi:10.3390/transplantology3020021_

Round 1

Reviewer 1 Report

This single-center study assessed the prognostic factor of unvaccinated solid ogan (liver, and kidney) recepient with COVID-19. Although the study is interesting, I have several concerns.

  1. Because multivariable analysis cannot be conducted due to instability, the present findings based on univariable analysis remains inconclusive.
  2. About the age issue, the present table 3 needs some modification to be interpretable and the time frame about figure 1 should be 90 days.
  3. Another major limitation was why these patients did not receive vaccination.
  4. Becasuse this study only included patients receiving liver and kidney transplanation, the present title using solid organ transplant was inappropriate.
  5. Please add the treatment policy against SARS-CoV-2 infections in both outpatients and inpatients setting in the study site.
  6. More comorbities, such as COPD, chronic liver disease, CKD are needed.

Author Response

The authors appreciate the Editors’ and Reviewers’ feedback. Please find below the point-to-point response to the reviewers’ comments.

Reviewer 1: This single-center study assessed the prognostic factor of unvaccinated solid organ (liver, and kidney) recipient with COVID-19. Although the study is interesting, I have several concerns.

1. Because multivariable analysis cannot be conducted due to instability, the present findings based on univariable analysis remains inconclusive.

Indeed, the small sample size limits the power of the statistical analysis, hence the instability; however, as this study includes all consecutive COVID-19 cases in eligible patients at our center and COVID-19 vaccination is now considered standard of care, the sample size is permanently limited. This does preclude meaningful multivariable analysis, although univariable analysis provides us with potentially targetable risk factors for mortality that can be investigated moving forward.

2. About the age issue, the present table 3 needs some modification to be interpretable and the time frame about figure 1 should be 90 days.

Thank you for your comment, both table 3 and figure 1 will be revised as suggested.

3. Another major limitation was why these patients did not receive vaccination.

This was as per study design: the study included consecutive solid organ transplant recipients who had contracted Covid-19 at the pre-Covid vaccination era: Patients in this study did not receive vaccination because the vaccine was not widely available in Arkansas at the time of study enrollment. The first vaccination was administered in Arkansas on December 14, 2020, over 10 months after start of study enrollment, and was initially made available to health care workers and first responders. Enrolled patients did not have the opportunity to be fully vaccinated until well after the end of data collection. Irrespective of the reason of non-vaccination, the authors believe that this cohort of prospectively populated registry of Covid-19 positive transplant recipients at the same institution, would offer valuable information as to the effect of the infection on nonvaccinated transplant population.

4. Because this study only included patients receiving liver and kidney transplantation, the present title using solid organ transplant was inappropriate.

Thank you for your comment, the title will be revised to reflect your suggestion. The title will read “SARS-CoV-2 Infection of Unvaccinated Liver and Kidney Transplant Recipients. A Single Center Experience of 103 Consecutive Cases.”

5. Please add the treatment policy against SARS-CoV-2 infections in both outpatients and inpatients setting in the study site.

We appreciate this recommendation. A discussion point with be added to the manuscript to reflect information below.

For outpatients, monoclonal antibody infusion treatment was recommended for symptomatic cases presenting as COVID-19 positive within 7 days of symptom onset. The first monoclonal antibody infusion was available December 2020 and a total of 21 patients enrolled in this study received this this treatment. For inpatients, patients with SpO2<94% on room air, RR>30, PaO2/FiO2 < 300mmHg, respiratory failure, shock, or multiorgan dysfunction were considered for treatment with Remdesivir and/or convalescent plasma. Administration was based on infectious disease clinician judgement and product availability. A total of 9 patients enrolled in this study received Remdesivir and 5 convalescent plasma.

Note that our treatment guidelines evolved during the study period, in compliance to the emerging scientific evidence, federal directives, and transplant bodies’ recommendations in the care of these patients.

6. More comorbities, such as COPD, chronic liver disease, CKD are needed.

Thank you for the opportunity to comment upon recipient comorbidities. Data was collected on multiple comorbidities to include HTN, Diabetes, Obesity, Cancer, Anemia, COPD, etc.… While zero patients in this study had COPD, comorbidities reported in table 1 can be expanded. CKD was not reported as a comorbidity, as by definition, all renal transplant recipients have CKD and reporting would not add value to the analysis. 

Once again, thank you for the opportunity to improve the quality of our manuscript and for your consideration for publication in Transplantology.

Emmanouil Giorgakis MD MSc FEBS FRCS

Asst. Prof. Surgery, HPB/Transplant

UAMS Medical Center

Little Rock, AR, United States

Reviewer 2 Report

The authors describe a large number of patients with SARS-CoV-2 infection in unvaccinated SOT recipients. These SOT recipients largely are renal transplant recipients, with some patients with liver transplantation or both renal and liver transplantation. The large number of patients is interesting for the transplant community, and the article is well written.

I have some minor comments / suggestions:

a) Line 58: here the recommendation of the ISHLT could be mentioned (Guidance from the ISHLT regarding the SARS-CoV-2 pandemic, revised February 1, 2021) which not only recommends withholding antimetabolites, but also mTOR inhibitors and azathioprine. 

b) Unfortunately the WHO severity of COVID was not documented, this could be interesting regarding the mortality of the patients. 

Author Response

The authors appreciate the Editors’ and Reviewers’ feedback. Please find below the point-to-point response to the reviewers’ comments.

Reviewer 2 :The authors describe a large number of patients with SARS-CoV-2 infection in unvaccinated SOT recipients. These SOT recipients largely are renal transplant recipients, with some patients with liver transplantation or both renal and liver transplantation. The large number of patients is interesting for the transplant community, and the article is well written.

The authors deeply appreciate the Reviewer’s kind response.

I have some minor comments / suggestions:

a) Line 58: here the recommendation of the ISHLT could be mentioned (Guidance from the ISHLT regarding the SARS-CoV-2 pandemic, revised February 1, 2021) which not only recommends withholding antimetabolites, but also mTOR inhibitors and azathioprine.

Thank you for the suggestion, commendation has been included in the revision. 

b) Unfortunately the WHO severity of COVID was not documented, this could be interesting regarding the mortality of the patients. 

Thank you for the suggestion, we will consider this variable for inclusion in future works. 

Once again, thank you for the opportunity to improve the quality of our manuscript and for your consideration for publication in Transplantology.

Emmanouil Giorgakis MD MSc FEBS FRCS

Asst. Prof. Surgery, HPB/Transplant

UAMS Medical Center

Little Rock, AR, United States

Reviewer 3 Report

This study aimed to study the SARS-CoV-2 specific mortality and associated risk factors of a cohort of 103 consecutive unvaccinated solid organ transplant recipients transplanted at a single academic transplant center, using a prospectively populated institutional SARS-CoV-2 transplant registry.          

Please kindly see the comments:

1. Apart from the age, did the author also investigate other factors that affect the death rate?

2. The reference format should be consistent

3. More data is suggested to support the conclusion that the age is the most important factor that affect the death rate

Author Response

The authors appreciate the Editors’ and Reviewers’ feedback. Please find below the point-to-point response to the reviewers’ comments.

Reviewer 3: This study aimed to study the SARS-CoV-2 specific mortality and associated risk factors of a cohort of 103 consecutive unvaccinated solid organ transplant recipients transplanted at a single academic transplant center, using a prospectively populated institutional SARS-CoV-2 transplant registry. Please kindly see the comments:

1. Apart from the age, did the author also investigate other factors that affect the death rate?

Indeed, we did (table 2). Recipient comorbidities did not elicit significant signal (data not shown). However, this can due to the well-known limitations of an inherently underpowered prospective registry analysis.

2. The reference format should be consistent

Thank you for the comment. The reference format has now been adjusted.

3. More data is suggested to support the conclusion that the age is the most important factor that affect the death rate

Thank you for the comment. As stated above, our cohort was powered to demonstrate only age group as survival factor; larger databases would possibly identify more risk factors in this patient population.

Once again, thank you for the opportunity to improve the quality of our manuscript and for your consideration for publication in Transplantology.

Emmanouil Giorgakis MD MSc FEBS FRCS

Asst. Prof. Surgery, HPB/Transplant

UAMS Medical Center

Little Rock, AR, United States

This manuscript is a resubmission of an earlier submission. The following is a list of the peer review reports and author responses from that submission.